# Person-Centeredness in Digital Primary Healthcare Services—A Scoping Review

**DOI:** 10.3390/healthcare11091296

**Published:** 2023-05-01

**Authors:** Ann-Chatrin Linqvist Leonardsen, Carina Bååth, Ann Karin Helgesen, Vigdis Abrahamsen Grøndahl, Camilla Hardeland

**Affiliations:** 1Faculty of Health, Welfare and Organization, Ostfold University College, P.O. Box 700, 1757 Halden, Norway; 2Department of Anesthesia, Ostfold Hospital Trust, P.O. Box 300, 1714 Grålum, Norway; 3Faculty of Health, Science, and Technology, Department of Health Sciences, Karlstad University, SE-651 88 Karlstad, Sweden

**Keywords:** digital health care, patient centeredness, person centeredness, primary health care, scoping review

## Abstract

**Background:** Transformation toward digital services offers unique opportunities to meet the challenges of responding to changing public healthcare needs and health workforce shortages. There is a knowledge gap regarding digital health and person or patient-centered care. **Aim:** The aim of the current scoping review was to obtain an overview of existing research on person or patient centeredness in digital primary healthcare services. **Design:** A scoping review following the five stages by Arksey and O’Malley. **Methods:** Literature searches were conducted in the databases PubMed, Scopus (Elsevier), APA PsychInfo (Ovid), Embase (Ovid), Cinahl (Ovid) and Cochrane Library in June 2022. The Preferred Reporting Items for Systematic reviews and Meta-Analyzes extension for Scoping Reviews (PRISMA-ScR) Checklist was followed. **Results:** The electronic database searches identified 782 references. A total of 116 references were assessed in full text against the inclusion and exclusion criteria. Finally, a total of 12 references were included. The included papers represent research from 2015 to 2021 and were conducted in eight different countries, within a variety of populations, settings and digital solutions. Four themes providing knowledge about current research on patient or person centeredness and digital primary health care were identified: ‘Accessibility’, ‘Self-management’, ‘Digitalization at odds with patient centeredness’ and ‘Situation awareness’. The review underlines the need for further research on these issues.

## 1. Background

Transformation toward digital services offers unique opportunities to strengthen healthcare systems and meet the challenges of responding to changing public healthcare needs with an increasing number of older people and people with complex diseases, as well as health workforce shortages [1,2]. Due to the different perspectives of academia, scientific institutions, industry and patients, a comprehensive definition of digital health is lacking. The World Health Organization (WHO) defines digital health as “the use of digital technologies in activities related to health” [3]. A 2020 systematic review identified the following components of digital health: e-health, m-health, telehealth and telemedicine, public health surveillance, personalized medicine, health promotion strategies, self-tracking, wearable devices and sensors, genomics, medical imaging and information systems [4]. Fatehi et al. [5] identified 95 unique definitions of digital health. Common to these definitions was a digital intervention that explored the provision of health care rather than the use of the technology itself. Moreover, the wellbeing of people, both at population and individual levels, was more emphasized than the care of patients suffering from diseases [5]. 

Both Dorning [6] and Ibrahim et al. [7] have explored quality in relation to digital health and healthcare services. As such, Dorning identified six quality domains, namely ‘patient safety’, ‘access to health care’, ‘effective treatment’, ‘efficient use of resources’, ‘equity of care across subgroups of populations’ and ‘person-centered care’ [6]. In addition, Ibrahim et al. [7] reviewed the evidence from review studies regarding digital health and different aspects of quality in medical care. Domains of quality being reviewed included effectiveness, accessibility, patient safety, efficiency, patient-centered care, and equity. There were considerable gaps concerning evidence on digital health for cost efficiency, equitable health care and patient-centered care [7]. The common aspect from Dorning [6] and Ibrahim et al. [7] is that person-centered/patient-centered care is assumed a quality domain in digital health and healthcare services.

Two different concepts are commonly used in care models that are not disease-centered, but models ensuring that people are involved in and central to the healthcare process as a key component in developing high-quality care, namely person- and patient centeredness [8]. The implementation of *person centeredness* has been stressed to require changes in norms and expectations for most healthcare systems [9]. Thus, developing person-centered care involves a shift in cultures [10] and is a complex task. In *patient-centered care*, an individual’s specific health needs and desired health outcomes provide the basis for healthcare decisions and quality measurements. As such, patients become partners with their healthcare providers [11]. Focus on patient-centeredness leads to improved patient satisfaction, better health, a reduction in the number of hospitalizations and re-hospitalizations and as economic benefits [12,13]. Godinho et al. [14] reviewed the literature on m-health implementation in Integrated People-Centered Health Services (IPCHSs). IPCHSs focus on empowering and engaging people, strengthening governance and accountability, reorienting the model of care, coordinating services within and across sectors, and creating an enabling environment [15]. The authors concluded that the use of m-health such as text messages, mobile apps, phone calls and video communication for people-centered care primarily have been used for community-based health issues [14]. The authors concluded that m-health can transition disease-centered services toward people-centered services. However, they also stated that there is limited evidence for large-scale implementation and international variation in reporting on m-health practice, modalities used and health domains addressed [14]. 

Even though patient-, people- or person-centered care have been researched for many years, these concepts still remain hard to operationalize, and there are challenges and barriers to putting them into practice [16,17]. Due to the terms ‘patient-‘, ‘people-‘ or ‘person-centered’ being interchangeably and extensively used without clear definitions, clarification or consensus [8], and all with the abbreviation PC, this abbreviation will be used further. 

The transfer of medical treatment and care from hospitals to primary health care has been a key idea in high-income countries to utilize healthcare resources in a sustainable way [18,19]. The WHO states that an impressive trend in national policies for digital health reflects the firm commitment to use digital technologies to advance the Sustainable Development Goals, support universal health coverage and shape the future of primary health care [20]. The Nordic countries have emphasized the need to strengthen digitalization in health care; however, patients’ and professionals’ perspectives require reinforcement [21,22]. There is a knowledge gap regarding digital health and PC care [7]. Knowledge about PC in digital primary healthcare services may be used to support the implementation of digital health in primary care [23]. Hence, the objective of the current scoping review was to obtain an overview of existing research on PC in digital primary healthcare services. 

## 2. Methods

### 2.1. Design 

To explore the research on PC in digital primary healthcare services, a broad range of research designs were of interest; therefore, a scoping review was chosen. In this study, the scoping review process followed the first five stages by Arksey and O’Malley [24], expanded upon by Levac et al. [25]: (1) identifying the research question, (2) finding relevant studies, (3) study selection, (4) charting the data, (5) collating, summarizing and reporting the results [24]. The study adheres to the Preferred Reporting Items for Systematic reviews and Meta-Analyzes extension for Scoping Reviews (PRISMA-ScR) Checklist [26]. 

### 2.2. Stage One: Identifying the Research Question

In order to obtain an overview of research on PC in digital primary healthcare services, the following research questions were chosen: How is research conducted within the field of PC and digital primary health care?What sort of available evidence exists within the field of PC and digital primary health care?What are the knowledge gaps within the field of PC and digital primary health care?

### 2.3. Stage Two: Finding Relevant Studies

The search strategies were developed based on the search words presented in Table 1. 

Medical subject headings (MeSHs) and search words described in Table 1 were used with the Boolean operator OR, and combined using the Boolean operator AND. 

The searches were conducted in the following six databases: PubMed, Scopus (Elsevier), APA PsychInfo (Ovid), Embase (Ovid), Cinahl (Ovid) and Cochrane Library in June 2022. The research team, with the help of a librarian who has expert knowledge of database searches, performed the database searches. References were handled using the EndNote X8 and Rayyan QCRI software [27]. Rayyan was used to structure the screening and extraction process. This tool also gave an opportunity to blind researchers when marking articles as either included, excluded, or undecided in relation to inclusion and exclusion criteria. 

### 2.4. Stage Three: Study Selection

Study selection was based on the following inclusion and exclusion criteria: 

Inclusion criteria:Primary/community healthcare services;Digital healthcare services;Person or patient centeredness;Middle-aged and aged adults (45 years and older) [28];Scandinavian and English language;Peer-reviewed articles;Not limited to study design or methodological approach.

Exclusion criteria:Research published before 2012;Conference abstracts;Unpublished material;Dissertations;Reviews;Electronic health records;Abstract missing.

Figure 1, a modified PRISMA flow diagram, shows the selection process. The electronic database searches identified 782 references. Duplicates were removed, which resulted in 700 references. These references were divided into two halves. Two of the authors, independently and blinded to each other, screened the titles and abstracts according to their relevance to ensure that the eligibility criteria were met. Decisions from the screening process labeled ‘undecided’ and where the two authors disagreed (conflicts) were discussed between all authors until an agreement was reached. As a result, 584 records were rejected. A total of 116 references were assessed in full text by two of the authors, blinded to each other, against the inclusion and exclusion criteria, which led to 104 excluded references and a total of 12 included references.

### 2.5. Stage Four: Charting the Data

A data-charting form was jointly developed by all of the reviewers to determine which variables to extract [26]. Two reviewers independently charted the data, discussed the results and continuously updated the data-charting form in an iterative process. 

The data-charting form included the following topics: first author, year, country, title, aim/research question, study design, data collection, analysis, setting, a sample including participants’ characteristics, summary of relevant findings and further research needs stated (Appendix A). The completed data-charting form was approved by all authors. 

### 2.6. Stage Five: Collating, Summarizing and Reporting the Results

In the final stage, the results sections in each paper were collated and coded by one of the authors (CH), and then reviewed and discussed by another author. The codes were then sorted into themes and patterns [24], through discussions between three of the authors (CH, VAG and CB), and discussed between all authors until a consensus was reached. Then the themes were written out in the text by the first author (ACLL). The results report on the current knowledge within the field of PC and digital primary health care. 

No formal methodological quality assessment (risk of bias) of the included articles was conducted, in line with guidelines for conducting a scoping review [29,30]. 

## 3. Results

### 3.1. Description of the Included Studies

The included papers (n = 12) represent research from 2015 to 2021, hence no relevant articles fulfilling the inclusion and exclusion criteria between 2012 and 2015 were identified. The studies were conducted in eight different countries (US, n = four; Sweden, n = two; and one paper from France, Iceland, Netherlands, New Zealand, Norway and the UK, respectively). Six of the papers use the term patient-centered [31,32,33,34,35,36], and six of the papers used the term person-centered [37,38,39,40,41,42]. The studies included a total of 1391 patients (of which 256 were veterans, aged48–85 years), 56 registered nurses (of which 48 were female, aged 30 to 65 years), 33 coaches (no information about sex/age), 17 ‘healthcare staff’, one manager, one physician, one technician, one occupational therapist, one administrator, and a nurse/physiotherapist pair (no information about sex/age). Due to the studies reporting age as age groups, mean or range, respectively, it was not possible to calculate any means across studies. Not all studies reported sex distribution, but when adding sex distribution in studies reporting this (n = eight), 80 percent of the patients were male. Patients’ illnesses included Chronic Obstructive Pulmonary Disease (COPD), Chronic Heart Failure (CHF), diabetes and colorectal/pancreatic cancer. Papers including healthcare personnel specified that the digital solutions explored were tested or implemented in an older patient population. 

Digital solutions in the included studies consisted of telemedicine (structured telephone support), home telemonitoring, digital medicine, electronic transfer of health information and other kinds of e-health interventions, such as web-based digital platforms, applications, social and clinical networks and digital personal health plans. 

### 3.2. Methods Used in the Included Studies

Two of the included studies were randomized controlled trials [31,37] randomizing patients to either usual care or usual care combined with PC. Six of the included studies used qualitative interviews [35,38,39,40,41,43]. Of these, one of the studies used a grounded theory approach, based on face-to-face or telephone interviews [38]. One of the studies used focus group interviews [39], two studies used semi-structured individual interviews [40,43], and one study combined observations with semi-structured individual interviews and focus groups [41]. Moreover, methods of qualitative analysis varied between studies. Additionally, one of the studies used qualitative analysis of text generated from apps [35]. The four quantitative studies had an interventional approach with questionnaires/surveys used alongside a digital intervention [32,33,34,44]. Different questionnaires and survey solutions were used: The Kansas City Cardiomyopathy Questionnaire (KCCQ) and the Patient Health Questionnaire 9 [31], the Multidimensional Health Locus of Control (MHLC) and the Intrinsic Motivation Inventory (IMI) and the System Usability Scale (SUS) [32], data registered from sensors recording ingestion dates and times, daily step count, BP, weight [33] and self-developed questionnaires [37]. 

### 3.3. Existing Evidence

Through sorting of codes/patterns across the included papers, four themes were identified providing knowledge about current research on PC and digital primary health care, namely ‘Accessibility’, ‘Self-management’, ‘Digitalization at odds with patient centeredness’ and ‘Situation awareness’. 

#### 3.3.1. Accessibility

Seven of the studies focused on ‘accessibility’ in a wide format and linked this to PC [32,35,36,38,40,41,44]. Barenfeld et al. [38] linked PC to easier access to informal interaction with healthcare professionals, other than the formal meeting points. Moreover, patients reported that information in general was more accessible. This was somewhat transferable to Dhillon et al. [32] who found that health applications reduced the need to use different websites for managing health and searching for information, and Smaradottir et al. [41] who found that improved cross-sectional information increased PC. In addition, Klein et al. [44] found that veterans thought that the information provided in the digital solution was useful and would help them be more involved in their health care. Jóhannsdóttir et al. [40] on one side found that written or visual information, such as from videos, seemed better recalled than oral instructions and hence was more accessible to patients. However, they also claimed the solution to have reduced accessibility in some older patients who did not have access to a computer or were not able to use it. The latter was also supported by Radhakrishnan et al. [36]. Moreover, Kim et al. [35] found that coaches benefited from referring to users’ electronic health records archived in an application while implementing PC communication strategies. 

#### 3.3.2. Self-Management

Regarding self-management, Kim et al. [35] found that the digital solution supported users’ development of a sense of control, ultimately contributing to effective self-management. Both Barenfeld et al. [38] and Klein et al. [44] supported this, focusing on how the digital solution helped users be more involved in their health care. The users reported positive experiences both during and after the intervention, feeling welcomed and encouraged to take responsibility for their health. However, Radhakrishnan et al. [36] identified conflicting views on the effectiveness of telehealth for accomplishing patient self-management of chronic disease. Advantages included daily information, reminders and early identification of deterioration, which empowered patients. Disadvantages included user mistakes due to lack of knowledge and that patients felt less responsible to self-manage due to telehealth [36]. Der Cingel et al. [39] also found that nurses differ in what they think self-management is, varying from ‘client’s autonomy in making decisions’, to ‘the need to motivate clients to perform tasks in daily life themselves’.

#### 3.3.3. Digitalization at Odds with Patient-Centeredness

In contrast, several studies also concluded that digitalization may be at odds with PC. For example, Der Cingel et al. [39] concluded that starting a relationship with a client should not start with suggesting e-health and that conversations with emotional support must have face-to-face contact. Almost all nurses in their study emphasized the fact that knowing about e-health is one thing, but actually using e-health and discussing it with clients definitely is the next step. Moreover, nurses stated that they are not able to see a patient’s behavior in their natural environment when using digital solutions and they want to see for themselves how patients are doing. This was supported by Smaradottir et al. [41] who found that, at a distance, you cannot bond with the client, and by Radhakrishnan et al. [36] who found that telehealth was perceived to diminish the quality of communication between nurses and patients. Moreover, Barenfeld et al. [38] found that the digital interventions’ effect on PC depended on the patients’ condition. As such, patients seemed to reject the intervention when they felt healthy (not in need of services) and also when they felt too ill to look for help. 

#### 3.3.4. Situation Awareness

Three of the studies indicated that digitalization in primary health care led to situation awareness. For example, Kim et al. [35] found that coaches had evolved a high level of situational awareness of users’ daily routines and behavior change by utilizing the digital solution for PC communication. Barenfeld et al. [38] found that the digital intervention gave the patients the possibility to combine their own expertise with that of an expert. This again led to a better understanding of their situation, as well as available promotive and preventive options. Contrarily, Radhakrishnan et al. [43] found that physicians were frustrated because telehealth data lacked information about patients’ contexts and therefore could not replace a good nursing assessment.

### 3.4. Knowledge Gaps

Eight of the included studies presented various suggestions for further research. Ali et al. [37] pointed out a need to further explore which patient at what stage of a disease would benefit the most from digital interventions, to be able to ensure PC care. This was also supported by Bekelman et al. [31] (focusing on heart failure patients) and Jóhannsdóttir et al. [40] (focusing on cardiac patients in the Arctic.) Der Cingel et al. [39] requested more research on nurses and how to convince them to assess and use e-health in a PC way. Dhillon et al. [32] concluded that longer-term studies with larger patient populations are needed to confirm and quantify the long-term health effects of digital intervention. This was also underlined by Radhakrishnan et al. [36] (focusing on visualization techniques and patients’, nurses’ and physicians’ decision-making processes when informed by telehealth data), Smaradottir et al. [41] (focusing on research on PC cross-organizational settings) and Kim et al. [35] (focusing on differences in coaching practices.) 

## 4. Discussion

This scoping review presents available evidence within the field of PC and digital primary health care. Only 12 studies from 2015–2021 were included, from a variety of countries, and focusing on a variety of populations, primary care settings and digital solutions. The methodology also varied between studies. Existing knowledge was collated in four themes, namely ‘Accessibility’, ‘Self-management’, ‘Digitalization at odds with patient centeredness’ and ‘Situation awareness’. Moreover, knowledge gaps stated in the included studies support the need for further research on PC in digital primary health care. 

Seven of the studies in the current scoping review linked PC in digital primary health care to the accessibility of information or interaction [32,35,36,38,40,41,42]. The World Health Organization has also emphasized the link between accessible information and PC care [45]. Moreover, a 2022 scoping review of research on the needs and barriers of people with impairments related to the use of digital health solutions supports that accessible digital health care has an opportunity to foster health equity and achieve health promotion, prevention and self-care [46]. 

Three of the included studies found that the digital solution contributed to effective self-management [35,38,42]. A 2020 systematic review found that technology supported evolving independence, empowerment and patient participation across primary and secondary healthcare services [47]. Self-care, independence, empowerment and participation are all elements seen as essential for achieving PC health care [48,49]. In contrast, four of the included studies underlined that digitalization could also be at odds with PC in primary health care [38,39,41,43]. For example, several studies have underlined that digitalization cannot replace human-to-human contact in healthcare services [50,51,52].

Several studies have shown that the digitalization of health care has changed healthcare professionals’ roles and responsibilities [53,54]. For example, Jarva et al. [55] found that healthcare professionals’ perceptions of digital health competence focused on the ability to provide PC care by evaluating the need and possibilities for using digital health services combined with more traditional methods [55]. Studies have emphasized the need to develop and implement more evidence-based interventions to engage patients, enhance patient-provider communication and facilitate shared decision-making to improve PC health outcomes in relation to digital healthcare services [47,50,56]. However, PC care remains conceptional in nature, leading to a disparity between how it is interpreted and operationalized within the healthcare system [57]. It has been addressed that a challenge to PC in care may be that the introduction and use of technology in care especially for older patients may lead to changes in treatment and health care. Health care supported by technology may be experienced as simultaneously supportive and contributing to an increased distance between persons [58], which is also indicated in the results of the current scoping review. Even if a digital health intervention is high quality, well-publicized and promoted, and patients and users are aware of and supported to sign up to it, there is no guarantee that they will register for it as other factors can affect their ability to enroll. In particular, busy lifestyles with competing demands on individuals for their time and commitment often take precedence over personal health [59]. A 2023 systematic review including 287 papers and analyzing the changes taking place in the field of health care due to digital transformation concluded that there is a great need for research on the implications of digitalization by different stakeholders [51]. 

The research group consisted of researchers experienced with conducting reviews and four of the researchers are also experienced with primary healthcare research. Moreover, the literature search was supervised by an experienced librarian, which together supports the validity of this scoping review. The inclusion and exclusion criteria may have limited our results. For example, including studies only focusing on middle-aged and aged adults (45 years old and older) may have excluded relevant studies. In addition, less restricted inclusion and exclusion criteria could potentially yield a higher number of included studies, providing a richer data set. Additionally, searching gray data and including searches in the reference lists of the included papers could have resulted in the identification of further articles. 

Due to not conducting any quality assessment of included papers, the trustworthiness or generalizability of the study findings cannot be stated. The range of methodology and approaches in the included studies support the choice of not conducting a systematic review. 

## 5. Conclusions

Previous studies have identified areas that are related to PC in digital primary health care, including that digitalization may increase the accessibility of services, patients’ ability for self-management and healthcare personnel’s situation awareness. However, studies also indicate that digitalization in primary health care may be at odds with PC health care. Based on the variability of study methods, settings, populations and digital solutions included in earlier research, we claim that further research is needed to extend the knowledge base regarding PC and digital primary health care. 

## Figures and Tables

**Figure 1 healthcare-11-01296-f001:**
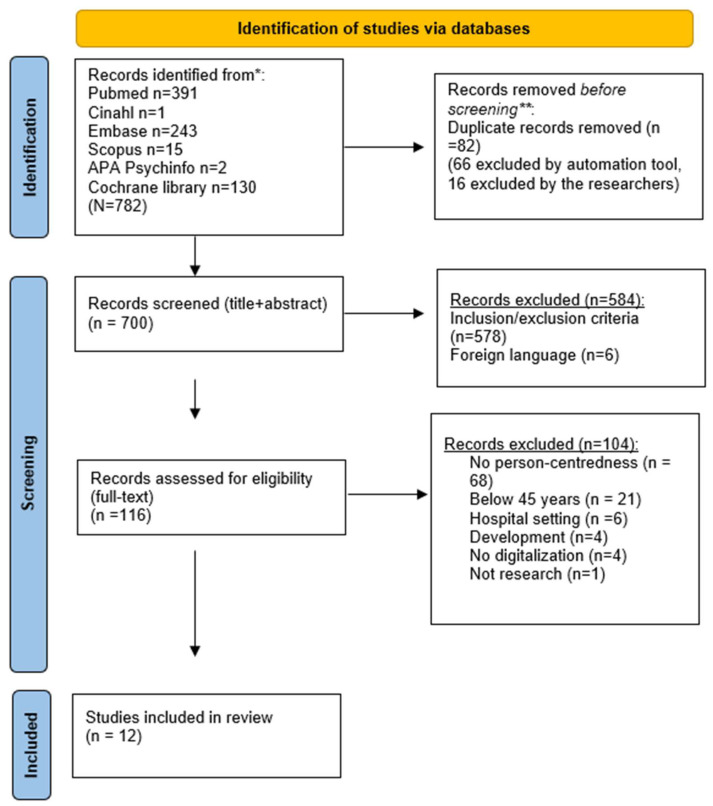
* Consider, if feasible to do so, reporting the number of records identified from each database or register searched (rather than the total number across all databases/registers). ** If automation tools were used, indicate how many records were excluded by a human and how many were excluded by automation tools. *From:* Page MJ, McKenzie JE, Bossuyt PM, Boutron I, Hoffmann TC, Mulrow CD, et al. The PRISMA 2020 statement: an updated guideline for reporting systematic reviews. BMJ 2021; 372: n71. doi: 10.1136/bmj.n71. For more information, visit: http://www.prisma-statement.org/ (accessed on 16 October 2022).

**Table 1 healthcare-11-01296-t001:** Search words.

Search Words.	
Primary health care OR Community health care AND Digital OR Digit * OR Digital Health OR E-health OR Technology enabled care OR Telecare OR Telemedicine OR Telehealth OR Computer-based technologies	Patient-centeredness OR Patient-centeredness OR Patient-cent * OR Patient cent * OR Person-centeredness OR Person-centeredness OR Person-cent * OR Person cent *
Columns 1 and 2 combined with AND.	

* truncation to identify terms starting with the present.

## Data Availability

Materials are available from the corresponding author upon request.

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
