# Peer review of "Person-Centeredness in Digital Primary Healthcare Services—A Scoping Review"

_healthcare, 2023, doi:10.3390/healthcare11091296_

Round 1

Reviewer 1 Report

This paper leaves me with more questions than answers. The conclusion of the article is weakly written and needs revision. The statement that ends with "...that is it not possible to conclude regarding PC and digital primary healthcare" is thoroughly incomplete and is poorly written leaving the reader of the article scratching their head and wanting to know what exactly can't be concluded?  The results section could be more developed for points 3.2, 3.3 and 3.4. In contrast to the sections that come after 3.4 and before 3.2 which are all solid paragraphs rather than one to two sentences. There doesn't need to be a 4.1 marker.  I think these minor corrections/clarifications would make the article stronger.

The English overall is proficient. Be careful when writing scoping that you do not write scooping which occurs in 2.1 line 2

Author Response

Reviewer 1

Comment 1: This paper leaves me with more questions than answers. The conclusion of the article is weakly written and needs revision. The statement that ends with "...that is it not possible to conclude regarding PC and digital primary healthcare" is thoroughly incomplete and is poorly written leaving the reader of the article scratching their head and wanting to know what exactly can't be concluded? 

Response: We thank the reviewer for this input, and have consequently revised the conclusion. However, we would like to underline that this is a scoping review, the primary aim to identify the knowledge-base and the knowledge-gap (please see Arksey & O’Malley, 2005: https://doi.org/10.1080/1364557032000119616). Please see manuscript with track changes.

Comment 2: The results section could be more developed for points 3.2, 3.3 and 3.4. In contrast to the sections that come after 3.4 and before 3.2 which are all solid paragraphs rather than one to two sentences.

Response: We thank you for the opportunity to revise the results section. The point 3.2 (Methods used in the included studies) has been more developed. In the original manuscript, heading numbers were not included, and has been added by the journal. Unfortunately, there has been a misunderstanding about the heading levels, and point 3.3 (Existing evidence) is only an introduction to the themes identified through analysis. The correct heading level is the following:

3.1. Description of the included studies

3.2. Methods used in the included studies

3.3 Existing evidence.

3.3.1 Accessibility

3.3.2. Self-management

3.3.3 Digitalization at odds with person-centredness

3.3.4 Situation awareness

3.4. Knowledge gaps.

We hope the reviewer find our revisions sufficient. Please see manuscript with track changes.

Comment 3: There doesn't need to be a 4.1 marker. 

Response: The heading has been removed.

Comment 5: I think these minor corrections/clarifications would make the article stronger.

Response: We thank the reviewer for acknowledging our manuscript, and for suggestions that we agree have made the article stronger.

Comment 6: The English overall is proficient. Be careful when writing scoping that you do not write scooping which occurs in 2.1 line 2

Response: We thank for this correction, and have of course revised throughout

Reviewer 2 Report

Transformation towards digital services offers unique opportunities to meet the challenges of responding to the changing public health care needs and health workforce shortages. There is a knowledge gap regarding digital health and person or patient centred care.

The authors proposed scoping review to get an overview of existing research on person or patient centredness in digital primary health care services.

They detected four themes providing knowledge about current research on patient or person centredness and digital primary healthcare were identified: ‘Accessibility’, ‘Self-management’, ‘Digitalization at odds with patient-centredness’, and ‘Situation awareness’. The review underlines the need for further research on these issues.

The study is interesting.

I have some minor suggestions with a pure academic spirit

1.      The abstract must better summarize the sections. For example the first sentence is not an aim

2.      Better detail the purpose, using for example the sub aims with bullet points.

3.      Avoid short paragraphs. Such as paragraph 3.2

4.      References must be cited with []

5.      Avoid the use of we and expand the conclusions

Author Response

Reviewer 2

Comment 1: Transformation towards digital services offers unique opportunities to meet the challenges of responding to the changing public health care needs and health workforce shortages. There is a knowledge gap regarding digital health and person or patient centred care. The authors proposed scoping review to get an overview of existing research on person or patient centredness in digital primary health care services. They detected four themes providing knowledge about current research on patient or person centredness and digital primary healthcare were identified: ‘Accessibility’, ‘Self-management’, ‘Digitalization at odds with patient-centredness’, and ‘Situation awareness’. The review underlines the need for further research on these issues. The study is interesting.

Response: We thank the reviewer for acknowledging our study.

Comment 2: I have some minor suggestions with a pure academic spirit

  1. The abstract must better summarize the sections. For example the first sentence is not an aim

Response: We thank the reviewer for this input, and have revised the abstract so that is better summarizes the sections. Please see manuscript with track changes as well as the clean copy.

Comment 3: 2.      Better detail the purpose, using for example the sub aims with bullet points.

Response: We thank the reviewer for this suggestion. However, in-line with Arksey & O’Malley (2005), identifying the research questions is step one in the methodology. Hence, the research questions presented under the methods section would overlap with potential sub aims. We hope the reviewer agree on this. Otherwise, we will of course revise in a next review round.

Comment 4.: 3.      Avoid short paragraphs. Such as paragraph 3.2

Response: We thank for this input, which was also stated by reviewer 1. We have consequently revised the methods section, and points 3.2 to 3.5.

Comment 5: 4.      References must be cited with []

Response: We thank for this input. This has of course been revised.  

Comment 6: 5.      Avoid the use of we and expand the conclusions

Response: We thank for this input, and have removed the “we” references in the manuscript throughout. The conclusion has been expanded/revised. Please see manuscript with track changes.

Reviewer 3 Report

This paper presents an interesting study on the concept of person-centredness and related publications. The study has a clear goal, and results and discussion sections are well developed, highlighting main outcomes and current limitations of the research. 

Author Response

Reviewer 3

Comment: This paper presents an interesting study on the concept of person-centredness and related publications. The study has a clear goal, and results and discussion sections are well developed, highlighting main outcomes and current limitations of the research.

Response: We thank the reviewer for acknowledging our manuscript.

Reviewer 4 Report

This article deals with an overview of existing research on patient centredness in digital primary health care. Literature searches have been conducted in databases such as PubMed, Scopus, and others. The method used including search words and Boolean operators is explained, although the final number (12) seems too low compared with the initial figure (700). The discussion results are not conclusive and it is required further research in order to establish a relation between patient centredness and digital primary healthcare. The method and discussion is correct, although some minor editing of english is required. For instance, correction is required in the sentence "... indicate that is it not possible to conclude regarding PC ...", and "often taken precedence over personal health". Also the word "mhealth" should unify the spelling (mhealth, mHealth...).

Correction is required in the sentence "... indicate that is it not possible to conclude regarding PC ...", as well as "often taken precedence over personal health". Also the word "mhealth" should unify the spelling (mhealth, mHealth...).

Author Response

Reviewer 4

Comment 1: This article deals with an overview of existing research on patient centredness in digital primary health care. Literature searches have been conducted in databases such as PubMed, Scopus, and others. The method used including search words and Boolean operators is explained, although the final number (12) seems too low compared with the initial figure (700).

Response: We thank the reviewer for this input. The number of included articles is based on the systematic approach described in the manuscript, and hence cannot be increased. However, we recognize that a higher number of included studies could have provided a more rich dataset, and we have added information about this in the limitation section. Please see manuscript with track changes.  

Comment 2: The discussion results are not conclusive and it is required further research in order to establish a relation between patient centredness and digital primary healthcare.

Response: We agree with this comment, and have revised the conclusion to better adhere with the results and discussion presented.  

Comment 3: The method and discussion is correct, although some minor editing of english is required. For instance, correction is required in the sentence "... indicate that is it not possible to conclude regarding PC ...", and "often taken precedence over personal health". Also, the word "mhealth" should unify the spelling (mhealth, mHealth...).

Response: We thank the reviewer for this input, and have thoroughly proof-read the manuscript for English editing.
